# Cuffless Blood Pressure Monitoring: Academic Insights and Perspectives Analysis

**DOI:** 10.3390/mi13081225

**Published:** 2022-07-30

**Authors:** Shiyun Li, Can Zhang, Zhirui Xu, Lihua Liang, Ye Tian, Long Li, Huaping Wu, Sheng Zhong

**Affiliations:** 1College of Mechanical Engineering, Zhejiang University of Technology, Hangzhou 310023, China; lishiyun@zjut.edu.cn (S.L.); 2111802265@zjut.edu.cn (C.Z.); 2112102048@zjut.edu.cn (Z.X.); lianglihua@zjut.edu.cn (L.L.); zsheng0012@163.com (S.Z.); 2Key Laboratory of Special Purpose Equipment and Advanced Processing Technology, Ministry of Education and Zhejiang Province, Zhejiang University of Technology, Hangzhou 310023, China; 3State Key Laboratory of Nonlinear Mechanics (LNM), Institute of Mechanics, Chinese Academy of Sciences, Beijing 100190, China

**Keywords:** blood pressure monitoring, cuffless, pulse wave, pulse transit time

## Abstract

In recent decades, cuffless blood pressure monitoring technology has been a point of research in the field of health monitoring and public media. Based on the web of science database, this paper evaluated the publications in the field from 1990 to 2020 using bibliometric analysis, described the developments in recent years, and presented future research prospects in the field. Through the comparative analysis of keywords, citations, H-index, journals, research institutions, national authors and reviews, this paper identified research hotspots and future research trends in the field of cuffless blood pressure monitoring. From the results of the bibliometric analysis, innovative methods such as machine learning technologies related to pulse transmit time and pulse wave analysis have been widely applied in blood pressure monitoring. The 2091 articles related to cuffless blood pressure monitoring technology were published in 1131 journals. In the future, improving the accuracy of monitoring to meet the international medical blood pressure standards, and achieving portability and miniaturization will remain the development goals of cuffless blood pressure measurement technology. The application of flexible electronics and machine learning strategy in the field will be two major development directions to guide the practical applications of cuffless blood pressure monitoring technology.

## 1. Introduction

Blood pressure, the pressure of the blood circulating through the body in the arteries, is an extremely critical parameter for human health [1]. Figure 1 shows the systolic and diastolic blood pressures. Systolic and diastolic blood pressure can reflect the condition of the body. Abnormal hypertension is one of the major risk factors of cardiovascular disease (CVD) [2]. With the aging of society, the increasing hypertensive population brings a potential high risk of CVD. Due to the shortcomings of traditional blood pressure monitoring equipment, such as large size, low comfort and no continuous measurement [3,4,5], it is particularly important to develop cuffless blood pressure monitoring technology.

The first physiological study of blood pressure monitoring techniques dates back to the 16th century, and William Harvey attributed the arterial pulse generation, paving the way for the study of modern pulse waves [6]. In 1905, Korotkoff first introduced the blood pressure measuring auscultation method, which included a cuff and stethoscope [7]. The auscultation method is still widely used in hospitals or clinics in today’s society. Another common method is the oscillometric method, which measures arterial blood pressure by inflating and venting a cuff with a built-in pressure sensor [8]. For a long time, the traditional auscultation and oscillometric method has been the clinical standard. However, as revealed in recent years, these measurement techniques can only perform occasional measurements, which is inconvenient and makes it difficult to meet the requirements of the continuous and comfortable monitoring of blood pressure. Many devices such as Somnotouch-NIBP have been clinically validated. Therefore, in recent decades, a cuffless, easy-to-use, continuous blood pressure monitoring technology has been proposed. Current studies in this field mainly focus on monitoring blood pressure through the analysis of physiological signals such as pulse wave [9,10,11,12,13,14,15,16,17,18,19]. Among all cuffless blood pressure monitoring approaches, pulse transit time (PTT) and pulse wave analysis (PWA) have received extensive attention and investigation.

PTT, the time at which a pressure wave is transmitted between two arterial sites, is a reliable and ubiquitous blood pressure monitoring method [20], usually approximated by pulse arrival time (PAT) and pulse wave velocity (PWV) [21]. Figure 2 shows the principle of PTT. The PAT is defined as the time interval between the electrical activation of the heart and arrival of a pulse wave at a location on the body, such as the fingers, toes, and forehead. The PWV is determined as the velocity at which the arterial pulse travels from proximal to distal in an artery.

In 2000, Chen et al. measured systolic blood pressure (SBP) with PAT determined by an ECG reference point and the pulse wave measured by a finger blood oxygen sensor. Due to the merit of this method, the error between the estimated BP value and the reference BP value is 4–10%. In 2008, Hassan et al. derived a regression model for estimating SBP only based on the PTT method, which did not require calibration for every individual subject, where the PTT was determined by photoplethysmogram (PPG) and electrocardiogram (ECG) signals. Jing Liu et al. used a four-channel PPG signal system to collect the PPG waveform of different blood vessels and ECG to evaluate the PTT, and found that multi-wave-PPG were more accurate compared with previous methods.

**Figure 2 micromachines-13-01225-f002:**
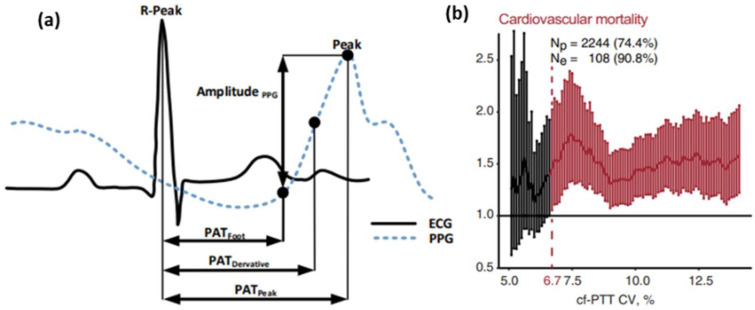
Schematic representation of PTT. (**a**) The characteristics of photoplethysmogram [22] and electrocardiogram. (**b**) The relationship between cardiovascular mortality and cf-PTT CV [23].

In the last few years, ultrasound-based methods have also been developed because ultrasound can accurately track the diameter waveform and wall thickness of the blood vessels. In 2013, ultrasound technology was employed to detect blood velocity to estimate BP and did not affect the normal activities of patients [24]. A balloon was placed over the wrist radial artery, and the collected ultrasound signal was adjusted by controlling the balloon pressure. In 2016, Joohyun et al. estimated ABP by employing a two-channel ultrasound system [25] and calculated PWV using a combination of the arterial vessel cross-sectional area and vascular elasticity to evaluate BP. This study may drive the development of new, readily available, inexpensive and powerful cardiovascular diagnostic tools in the future.

Basically, the PTT-based methods are performed under the assumption that the arteries are passive thin-walled and purely elastic tubes [26]. Nevertheless, pulse waves are influenced by various factors, and the vessel wall varies among individuals [27]. The BP-PTT relationship is a complex and nonlinear relationship [28]. Therefore, with the development of computer technology and data analysis tools, a PWA method for the reference signal processing and extraction was proposed for some features in pulse waveforms [29]. These features are commonly used to develop models to estimate blood pressure through machine learning. Several researchers have investigated the feasibility of wearable and continuous BP predictions using only one PPG sensor. The BP of four feature vectors was extracted from PPG signals highly correlated with BP and the diastolic time [30]. This study demonstrated that it is feasible to estimate blood pressure using the relaxation time of the photo plethysmography signal. 

Additionally, it was suggested that PTT and PWA can be combined to improve the accuracy of BP estimation. Xiao presented a new indicator for tracking low-frequency changes, the photohemogram intensity ratio (PIR) [31]. Lin et al. proposed the intensity ratio of the first derivative wave (1st dPIR) and a novel blood pressure estimation algorithm, which uses both PTT and 1st dPIR to enhance the accuracy of BP estimation. The 1st dPIR is defined as the PIR of the first derivative wave of PPG [32]. Ibrahim et al. measured the PTT from wrist to finger using biopotential-based impedance and PPG and compared it to the PTT measured by PPG from ECG to finger, proving that it can effectively replace the traditional PTT measurement strategies [33]. Liu et al. investigated the pressure pulse wave signals from a piezoelectric-induced sensor on the radial artery of the left wrist. Twenty-one features including PTT and PIR were extracted from pressure pulse wave (PPW) [34]. In the past few years, more than 30 features have been proposed, such as the PPW characteristic value K, peak amplitude of 1st dPPWr, etc. [35]. To meet the AAMI standard, many machining learning models have been employed, such as a support vector machine [36], random forest, feedforward neural network, etc. More advanced methods have also been proposed, such as recurrent neural networks and LSTM, which show great performance in long-term continuous monitoring [37,38]. In 2018, a hierarchical artificial neural network based on the waveform long-term–short-term memory(LSTM) blood pressure estimation model was proposed. The model consists of the following two hierarchical levels: the ANN level extracts the necessary morphological features from ECG and PPG waveforms, and the LSTM level accounts for the time domain variation of the features extracted from the ANN level [37]. The proposed model is able to automatically extract the necessary features and their time–domain variations to reliably estimate blood pressure in a cuffless and continuous manner. The Somnotouch-NIBP noninvasive continuous BP monitor carries out all validation criteria of ESH-IP 2010 both for SBP and for DBP levels [39]. Although the research of cuffless blood pressure monitoring based on pulse waves is still in its infancy, a number of review articles summarized the achievements in this field. These reviews were organized in terms of technical contents. In order to demonstrate the development of blood pressure monitoring and future research directions, the bibliometrics technology defined as “ a way to study the structure, quantitative relationship, change pattern and variable management of literature information by using mathematics, statistics and other measurement methods to explore the structure, characteristics and pattern of the literature system and bibliometric characteristics as the object of study” was applied to achieve an historical and comprehensive perspective in the area of cuffless blood pressure monitoring.

Quantitative methods such as mathematics [40] and statistics are utilized in bibliometrics to provide statements about the qualitative features of scientific research. Bibliometric methods can assess the productivity of institutions, countries, and authors to explore research hotspots/frontiers in specific fields [41,42]. Bibliometric techniques were proven to be an effective tool for analyzing research activities in various fields such as robotics [43], nanomaterials [44], and marketing [45]. We analyzed the field of cuffless blood pressure monitoring through the following aspects: (1) historical map of the topic; (2) the main contributors including countries, institutes, research groups, authors, and leading research areas; (3) cooperation patterns between countries, institutes, and authors; (4) the most productive journals; (5) top articles with highest citation number; and (6) research interests and perspectives.

## 2. Methods

The purpose of this paper was to highlight current developments in blood pressure measurement technologies, advanced research areas and institutions, and future trends, through a comprehensive review of articles published since 1990.

The collection of papers was achieved through the Web of Science (WoS) database. The WoS Core Collection includes the following three Citation Indexes: Science Citation Index Expanded (SCIE), Social Science Citation Index (SSCI) and Art and Humanity Citation Index (A&HCI), and more than 12,000 authoritative and influential international academic journals in the fields of natural sciences, engineering, biomedicine, social sciences, arts and humanities. The search strategy was used to retrieve articles that contain the following terms in their title, abstract or keywords: (“ cuffless” OR “non-inva*” OR “wearable” OR “ubiquitous measurement”) AND (“photoplethysmogram (PPG)” OR “pulse arrival time (PAT)” OR “pulse transit time”) AND “Blood pressure”.

As this review focused specifically on cuffless blood pressure monitoring, we identified abstracts and excluded some articles in certain categories. Since the literature searched in this paper was from the “topic” search of WoS, with mainly the titles, abstracts and keywords of the literature retrieved, some relevant literature may not be included. The impact factors for each journal mentioned in this work were determined by the 2019 journal citation report. Interventional studies involving animals or humans, and other studies requiring ethical approval, must list the authority providing the approval and the corresponding ethical approval code.

## 3. Results

Through the previously presented methods and data sources, Figure 3 shows the number of published literature related to cuffless non-invasive wearable blood pressure monitoring worldwide. It was observed that 83 countries have published 2091 publications, among which 11 are highly cited articles of Essential Science Indicators. As shown in Figure 3, the number of publications generally increased from 1990 to 2019. The annual growth rate is increasing, with more than 200 publications in the past three years, indicating that the research interest of scientists and engineers in this field has gradually increased in recent years.

### 3.1. Global Contributions and Leading Countries

The Bubble chart of the top 20 keywords by year is shown in Figure 4. Colors in all charts are distinguished according to the degree of keyword use. Since 2004, ‘‘blood pressure’’ has remained a widely researched keyword in scientific articles, especially in the years following 2012. The trend of publications using the keyword ‘‘blood pressure’’ is consistent with the polyline shown in Figure 3. After 2012, works with the keyword “photoplethysmography” were widely published, among which PTT and PWV represented the two main methods of blood pressure measurement.

The analysis of academic research results at the national or regional level can reveal which country/region is more valued in a certain discipline. From 1990 to 2020, 73 countries participated in cuffless blood pressure monitoring. Table 1 lists the top 20 most productive countries and regions in the field of cuffless blood pressure monitoring.

As shown in Table 1, the United States has the highest number of publications on cuffless blood pressure monitoring technology, with 405 articles and a total of 5300 citations. Followed by China and England, with a total of 296 articles and 284 articles published with total citations of 2260 and 4293, respectively. The top three countries in terms of average citations were England, Switzerland and the USA. Among the top ten countries or regions in terms of total publications, India has the lowest average citations. From the overall perspective of the top 20 countries, research on this technology is mainly concentrated in developed countries where medical care and personal attention are more advanced. As can be seen from Figure 5, cuffless BP monitoring has been a topic of interest in most countries over the past few years.

The collaborative relationship of the top 20 productive countries is displayed in Figure 6. The size of the nodes is proportional to the total number of items in each country. The lines represent collaboration between countries, and the thickness of the lines indicates the strength of cooperation. The top 20 countries/regions cooperate closely, with the United States being the most active partner with other countries, especially with China, the United Kingdom, Canada, and Australia. The reason may be that the United States is the top leader in this field, and researchers from various countries want to cooperate with the United States. The close cooperation between China and the United Kingdom, Japan, Germany and other countries indicates that the field of blood pressure monitoring is related to the health of people around the world, and that close communication and cooperation is an effective way by which to promote the development of this field.

### 3.2. Contribution of Leading Institutions

Detailed and specific information for scholars in the field of cuffless blood pressure monitoring is provided through the statistical analysis of the research institution. Table 2 summarizes the top 20 institutions in cuffless blood pressure monitoring research in terms of productivity, and their total number of publications, citations, and h-index. Apparently, most of them are from the top 20 productive countries, and the main institutions are also from the USA, the UK and China. The Chinese University of Hong Kong published 72 papers, with a total of 1305 citations [46,47] and an h-index of 16. The second institution is the Chinese Academy of Sciences in China, which has published 60 papers and been cited 902 times [48,49], with an h-index of 16. The third institution, the University of London in the United Kingdom, published 35 articles and received a total of 599 citations, with an h-index of 13. The top two published articles are both from China, and each article received a high citation, indicating that Chinese institutions have a strong research interest in this field and a high quality of research. Researchers at other institutions can seek cooperation with these institutions. National Chiao Tung University, Imperial College London and Macquarie University ranked the top three on average per article. To some extent, the quality of the articles published by these institutions in this field is relatively high, which has a greater impact on researchers worldwide.

### 3.3. Leading Journals in Terms of Number of Publications in Cuffless Blood Pressure Monitoring

It is essential for scholars and those interested in cuffless blood pressure monitoring to know of the journals that publish studies of cuffless blood pressure monitoring and decide which journal to focus on or submit to.

From 1990 to 2020, a total of 2091 papers were published in 1131 journals in related fields, and the top 20 journals are listed in Table 3. As shown in Table 3, *Physiological Measurement* (141), *Medical & Biological Engineering & Computing* (46), IEEE *Transactions on Biomedical Engineering* (44) are in the leading position. The publications of the above three journals accounted for 11.05% of the total publications. The number of publications in the top 20 journals presented in Table 3 totaled 502, accounting for 24%. The following three journals published more than 20 articles: *Medical Engineering & Physics*, *Annals of Biomedical Engineering*, and the 2017 39th annual international conference of the IEEE engineering in medicine and biology society (embc) which published 23, 22 and 20 articles. In terms of IF, IEEE *Transactions on Biomedical Engineering* and *Annals of Biomedical Engineering* are in the leading position. Above all, interest in cuffless blood pressure monitoring is apparent in a high number of journals.

### 3.4. Contribution of Leading Authors

Table 4 shows the top eight authors in terms of the number of articles published, and Figure 7 shows the number of articles published per year by the top ten authors. The Zhang group ranks first in the number of publications [50], followed by Mukkamala R [51] and Hahn [52]. Among the top ten authors of the most published papers, two are from the Chinese University of Hong Kong. The first author, Zhang, published 40 papers, accounting for 1.91% of the total publications, and maintained a very high average citation index, reaching 25.80 times per article, indicating that Zhang’s research group has a wide influence in this field. Mukkamala R ranked second, with 24 publications, accounting for 1.15% of the total publications. The average citation index was 19.63 times. The third position was Hahn, Jin-Oh, with 23 publications, accounting for 1.10% of the total publications, with an average of 19.91 citations per article.

We also found that Zhang’s group worked closely with Ding and Liu of the University of Chinese Academy of Sciences. This is mainly because Zhang worked at the University of Chinese Academy of Sciences after working at the University of Hong Kong. At the same time, many of the first eight authors collaborated more closely, indicating that researchers often communicate and cooperate, typically providing innovative developments in this field.

### 3.5. Analysis of the Most Cited Articles

Citation impact is widely accepted as an indicator for evaluating scientific papers, and Table 5 provides the top 10 most cited papers between 1990 and 2020.

Of the 2091 papers published from 1990 to 2020, 23 were cited more than 100 times, or at a rate of 1.10%. As shown in Table 5, the top 10 most-cited articles were published from 1996 to 2016. Most of the top ten papers were published in top journals, including Physiological Measurement, Advanced Materials, IEEE Transactions on Biomedical Engineering, etc. The most highly cited article is “Photoplethysmography and Its Application in Clinical Physiological Measurement”, published in Physiological Measurement in 2007 [53]. It was cited 1327 times in total, far exceeding the second most cites article, and TCY was cited 94.79 times, greatly influencing this field. A section of the article described that PPG can be used to monitor blood pressure. Although the origin of the PPG signal is not fully understood, it is closely related to cardiovascular parameters. At the same time, this article introduced that the PPG technology was applied to measure BP and blood oxygen, driven by the wide application of small semiconductor devices.

The sixth ranked article also mentioned PPG combined with wearable biosensors for the mobile monitoring of blood pressure. In addition, the fourth ranked article also used PPG to measure arterial response, which is indirect to changes in blood pressure. The 7th ranked article used the same approach.

It is worth noting that the second ranking article “Monitoring of Vital Signs with Flexible and Wearable Medical Devices” was published in 2016 with a TCY of 81 [54]. It mainly revealed the applicability of wearable sensors in health aspects such as heart, blood pressure and blood glucose under the development of wireless technology, low-power electronics and the internet of things. In addition, the essential components required for important sensors such as their manufacturing, sensor systems, power consumption and data processing were discussed in this review. The eighth and ninth ranked articles showed a wearable physiological monitoring system that can be used to measure blood pressure. The third-ranked article did not report on a specific blood pressure measurement technique, but rather on unremarkable sensing and wearable devices that have implications for the practical application of new blood pressure measurement technologies. The tenth-ranked article similarly offers a new idea about pressure sensors that measure blood pressure.

It is worth noting that the article “Toward Ubiquitous Blood Pressure Monitoring via Pulse Transit Time: Theory and Practice” published in 2015 ranked fifth with a TCY of 40.67 [20]. Compared with other top rankings, this paper introduced the theoretical models of PTT and BP, and reviewed the previous work from theory to practice. The prospective nature of this paper covered major studies in the field of cuffless blood pressure monitoring in recent years.

The top ten most cited articles originated from six countries, including the United Kingdom, the United States, China and Switzerland. To a certain extent, a range of countries have produced in-depth and extensive research in this field. The United States accounted for three articles of the top ten articles and has the most extensive influence in the area of cuffless blood pressure monitoring. China came in second with two.

### 3.6. Contribution of Leading Research Areas

Research areas reflect the application scopes of the topic. According to the collected data, the research on cuffless blood pressure monitoring includes 124 fields. Table 6 includes the top 20 research directions of cuffless blood pressure monitoring technology. The “Engineering, Biomedical” field ranked first, accounting for 37% of the total number of published articles, with a total citation frequency of 6485 times. Besides “Engineering, Biomedical”, only “Engineering, Electrical & Electronic” accounted for more than 10%, and “Computer Science, Interdisciplinary Applications” and “Medical Informatics” ranked third and fourth, respectively. In terms of ACPP indicators, Nanoscience & Nanotechnology, Automation & control systems, and Mathematical & Computational Biology, ranked the top three, with an ACPP of 23.88, 23.02, and 18.01, respectively. These three areas are closely related to flexible electronics and machine learning in cuffless blood pressure monitoring.

### 3.7. Analysis of Reviews for Each Year

The reviews on the topic of cuffless blood pressure monitoring are shown in Table 7, and the most related reviews and the summary of techniques are listed in Table 8. As listed in Table 7, the first review on cuffless blood pressure monitoring was published in 2006. Before 2014, there were only a few reviews related to this field. Since then, there have been at least two reviews published on this topic every year, and the number of review articles in this field has increased significantly, indicating that cuffless blood pressure monitoring has become a research hotspot. The multidisciplinary nature of cuffless blood pressure monitoring and the diversity of the authors’ professional backgrounds have resulted in the diversity of contents, analysis perspectives, and arguments of their publications. Despite the different perspectives, all the reviews are from technical standpoints, such as sensors, measuring systems, and signal processing.

Most reviews cover the latest work in this field around the world and summarize technical aspects as listed in Table 7 and Table 8. Reviewers consistently comment on the current issues and future work. These technical summaries, comments and recommendations are important for researchers and potential followers in this field.

Combining Table 7 and Table 8, it can be seen that almost all reviews related to cuffless blood pressure monitoring mentioned the method of estimating blood pressure based on PTT, which is consistent with the results shown in Figure 4. The PTT-based method is the most studied and mature technology in the field of cuffless continuous BP measurement [57]. The third column of Table 8 shows that PAT or PTT calculations mainly use PPG and ECG. Some researchers attempted to calculate PAT or PTT using BCG and PCG [58,59], while others tried to calculate them using double PPG sensors or pressure sensors. Overall, ECG and PPG are the most effective techniques for calculating PTT and PAT, which are not only convenient and accurate, but the ECG can reflect other healthy vital signs. 

Additionally, signal feature processing technology and BP models are widely mentioned. It is noteworthy that in the previous reviews, most BP models were based on PTT and linear regression methods. Then, non-linear and multivariable linear regression models were developed. After 2016, articles mentioning machine learning (SVM, BP network, random forest, etc.) and deep learning (Artificial Neural Network (ANN), Long Short-term Memory (LSTM) and Recurrent Neural Network (RNN), etc.) methods of pulse wave signals characterization emerged, indicating that non-invasive data processing technology and blood pressure monitoring have become increasingly related, which is crucial for the future development of this field [29,60,61]. Furthermore, the emergence of flexible electronics applications in this field was noted in a recent review [55,62]. The introduction of nanomaterials and MEMS technology to manufacture flexible sensor systems, such as electronic skins, flexible photoelectric sensors and flexible ultrasonic sensors, can effectively reduce the volume of cuffless blood pressure monitoring systems, preferably to obtain more reliable signals, and provide new ideas for the development of cuffless blood pressure monitoring technology [63,64]. The use of flexible pressure sensors [65], as shown in Table 8, based on PPW was investigated in previous years and has recently become a main focus in the field, because the development of ML and soft electronic methods based on PPW can enhance the accuracy required for BP monitoring.

## 4. Discussion

### 4.1. Research Interests and Perspectives

Based on the above aspects, in order to promote the practical applications of cuffless blood pressure monitoring technology, the simultaneous development of a sensor system and data analysis processing is an effective strategy. 

In the future, despite some studies producing questionable results [2,66], PTT/PAT will remain the base for BP measurement. PPG and ECG signals will still be the main signal sources for cuffless blood pressure monitoring and analysis [55,67]. However, with the development of PWA, the research on pressure pulse wave signals and ultrasonic pulse wave signals will be more concerned with offering BP measurements in a cuffless and wearable manner [36,56,68,69]. The application of this field will become an important research direction in the future. In the future, there is no doubt that machine learning will continue to develop. Automatic feature extraction is a vital issue, and combining time and frequency domain methods to improve long-term monitoring accuracy may also present a potential research direction. Furthermore, the process of machine learning in pressure pulse wave and ultrasonic pulse wave signals will become a new direction with the development of flexible electronics. The current cuffless blood pressure monitoring methods are basically performed on small homogeneous groups. The establishment of dynamic models of diverse groups of subjects will be very challenging.

### 4.2. Limitations

This article demonstrated the current development of cuffless blood pressure monitoring technology from the perspective of bibliometrics. However, the following limitations still exist.

First of all, the database in this article is only from the core set of WoS, and the comprehensiveness of the publications is insufficient; therefore, we did not cover the extensive research in this field, and some high-quality publications may have been overlooked. Secondly, this article only collected the development of this field from the perspective of bibliometrics, which cannot explore the underlying reasons behind the results presented in this paper. The focus of future work should be to collect more comprehensive information with a deeper understanding of the reasons that promote the development of cuffless blood pressure monitoring technology.

## 5. Conclusions

Although blood pressure measurement has represented a research topic for a number of years, cuffless blood pressure monitoring is still a relatively new research field. We highlighted the development of this field using bibliometrics based on the WoS core database using the increasing number of publications in recent years. Among all the regions, the United States, East Asia and Europe contributed most of the original research.

The institutions that published the most publications are also located in these countries. The first ranked institution is the Chinese University of Hong Kong, which published 72 articles with 1305 citations. The second and third institutions are the University of Chinese Academy of Sciences and the British University. Two of the top three institutions are from China. Combined with high-yield publications, China also ranks among the top, indicating that China attaches more importance to this field and has conducted more in-depth research.

According to the analysis, the top three most productive authors are Zhang, Mukkamala and Hahn. Two of the top eight authors are from the Chinese University of Hong Kong, and their total citations and average citations per article are also high. The other six authors among the top eight are from the top 20 countries in terms of the number of publications. In reference to the journals, 2091 articles were published in 1131 journals, which indicated that the topic is widely researched. According to the analysis of most cited articles and reviews on cuffless blood pressure monitoring, machine learning and flexible electronics were found to be the most attractive aspects to researchers.

In summary, the accuracy of current cuffless blood pressure monitoring is not yet acceptable in terms of medical standard requirements, but there are already clinically proven devices for use in clinical practice. It was also found that there is a mismatch between theory and current practice. While obstacles in developing cuffless blood pressure monitoring systems remain, it is very likely that accurate, wearable, reliable, comfortable and continuous blood pressure monitoring technology will become available in the near future.

## Figures and Tables

**Figure 1 micromachines-13-01225-f001:**
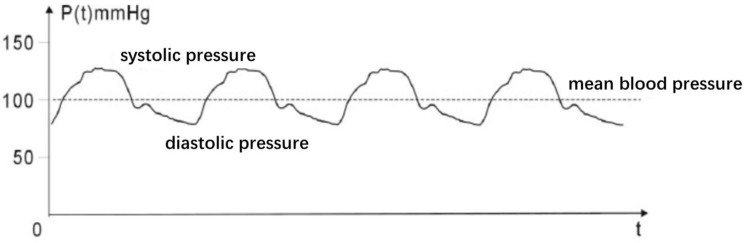
Schematic representation of the systolic and diastolic blood pressure.

**Figure 3 micromachines-13-01225-f003:**
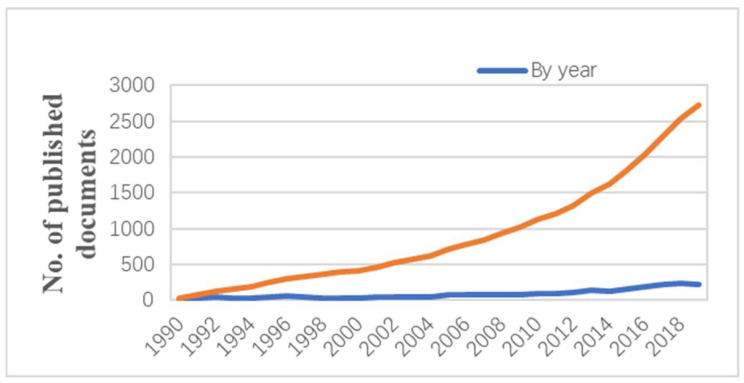
Number of published documents from the global related to cuffless blood pressure monitoring.

**Figure 4 micromachines-13-01225-f004:**
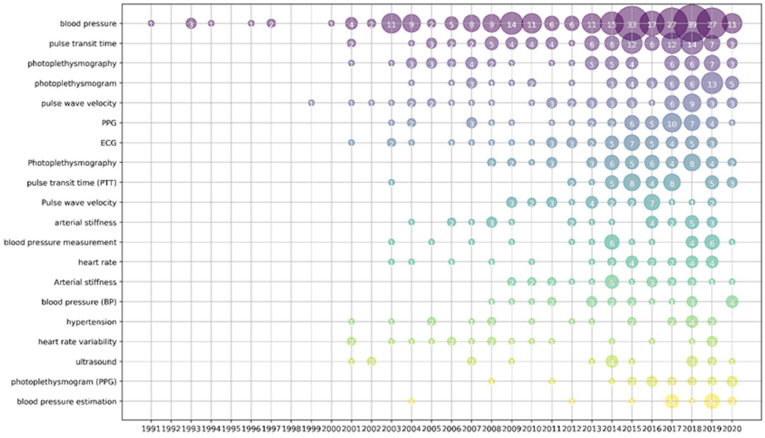
Bubble chart of top 20 keywords by year.

**Figure 5 micromachines-13-01225-f005:**
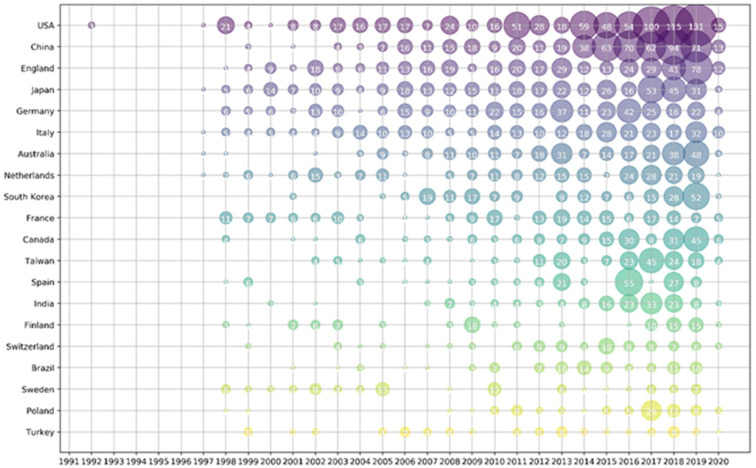
Bubble chart of top 20 productive countries/regions by year.

**Figure 6 micromachines-13-01225-f006:**
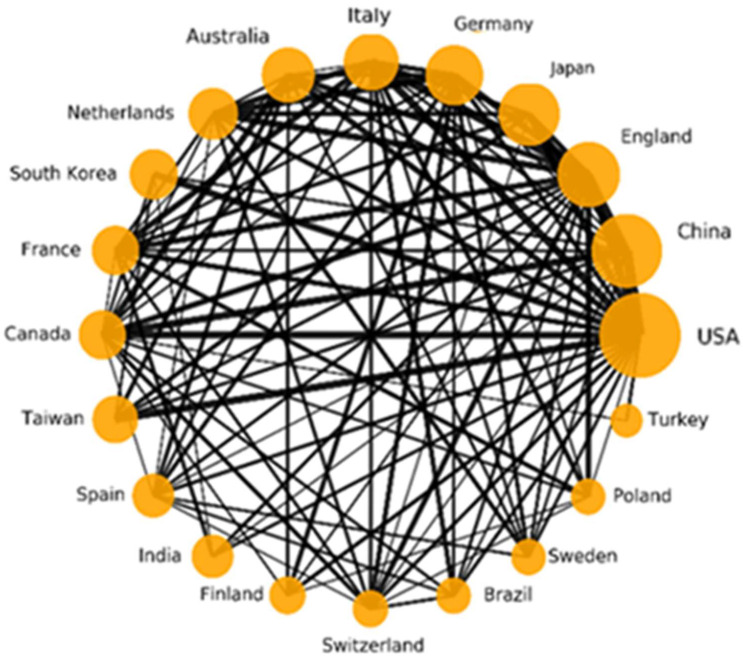
Collaboration matrix map among the top 20 most productive countries/regions.

**Figure 7 micromachines-13-01225-f007:**
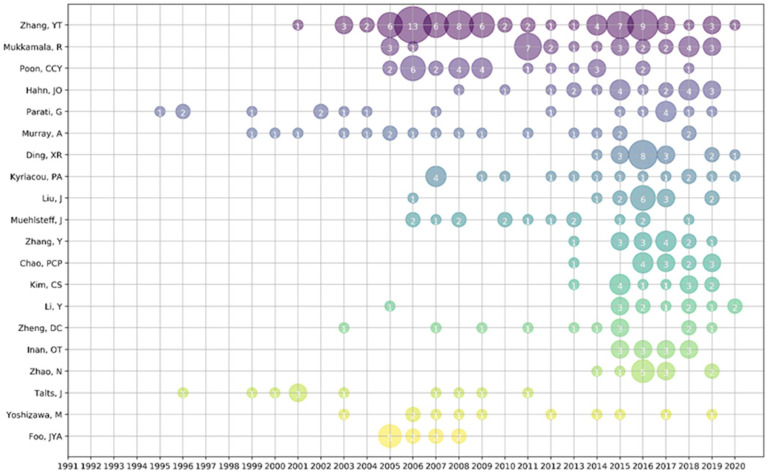
Top 10 productive authors in cuffless blood pressure research.

**Table 1 micromachines-13-01225-t001:** Top 20 most productive countries and regions in cuffless blood pressure monitoring field during 1990–2020.

Rank	Country/Region	TA	TC	ACPP	nCC
1	USA	405	5300	13.09	163
2	China	296	2260	7.63	107
3	England	284	4293	15.11	79
4	Japan	162	1356	8.37	24
5	India	113	440	3.89	11
6	Taiwan	107	486	4.54	25
7	Germany	105	1260	12.00	46
8	South Korea	104	659	6.33	24
9	Canada	104	1028	9.88	44
10	Italy	89	636	7.14	37
11	Australia	87	910	10.46	42
12	Hong Kong	75	1386	18.48	9
13	Netherlands	54	755	13.98	34
14	Switzerland	53	777	14.66	30
15	France	40	564	14.1	23
16	Spain	37	445	12.02	17
17	Brazil	35	270	7.71	9
18	Poland	35	123	3.51	9
19	Malaysia	33	71	2.15	11
20	Sweden	33	296	8.97	16

TA: total articles; TC: total citations; ACPP: average citations per publication; nCC: number of cooperative countries.

**Table 2 micromachines-13-01225-t002:** Top 20 most productive institutions of publications during 1990–2020.

Rank	Institutions	TA	TPR%	TC	ACPP	h-Index	Country
1	Chinese University of Hong Kong	72	35.93	1305	18.12	16	Hong Kong
2	Chinese Academy of Sciences	60	46.27	902	15.03	16	China Mainland
3	University of London	35	42.12	599	17.11	13	England
4	Massachusetts Institute of Technology (MIT)	34	36.12	989	29.09	13	USA
5	Harvard University	33	42.76	447	13.54	12	USA
6	Tohoku University	31	52.59	198	6.39	7	Japan
7	Shenzhen Institute of Advanced Technology, CAS	28	43.03	269	9.61	10	China Mainland
8	Michigan State University	27	38.30	514	19.04	9	USA
9	University of California System	27	36.28	852	31.56	12	USA
10	Indian Institute of Technology System (IIT System)	26	47.64	238	9.15	6	India
11	Tsinghua University	24	41.31	322	13.42	7	China Mainland
12	University of Maryland College Park	24	52.30	458	19.08	8	USA
13	Macquarie University	24	52.50	132	5.50	6	Australia
14	University System of Maryland	24	52.30	458	19.08	8	USA
15	Philips	24	48.80	136	5.66	6	Netherlands
16	Philips Research	22	50.55	125	5.68	6	Netherlands
17	University System of Georgia	21	46.83	438	20.85	8	USA
18	National Chiao Tung University	20	68.40	69	3.45	5	Taiwan
19	Georgia Institute of Technology	19	48.34	421	22.15	7	USA
20	Imperial College London	19	54.54	480	25.26	7	England

TPR, the percentage of articles of journals in total publication.

**Table 3 micromachines-13-01225-t003:** Top 20 journals publishing articles related to cuffless blood pressure.

Rank	Institutions	TA	TC	ACPY	IF
1	Physiological Measurement	141	3769	139.59	2.02
2	Medical & Biological Engineering & Computing	46	1098	93.48	1.88
3	IEEE Transactions on Biomedical Engineering	44	1398	97.73	4.32
4	Medical Engineering & Physics	23	359	86.96	1.70
5	Annals of Biomedical Engineering	22	390	95.45	3.38
6	2017 39th Annual International Conference of the IEEE Engineering in Medicine and Biology Society (EMBC)	20	45	60.00	n/a
7	2016 38th Annual International Conference of the IEEE Engineering in Medicine and Biology Society (EMBC)	19	54	84.21	n/a
8	2015 Computing in Cardiology Conference (CinC)	17	108	82.35	n/a
9	2005 27th Annual International Conference of the IEEE Engineering in Medicine and Biology Society, Vols 1–7	17	245	76.47	n/a
10	Biomedical Engineering-Biomedizinische Technik	16	14	25.00	0.89
11	2011 Annual International Conference of the IEEE Engineering in Medicine and Biology Society (EMBC)	15	43	73.33	n/a
12	2013 35th Annual International Conference of the IEEE Engineering in Medicine and Biology Society (EMBC)	15	99	100.00	n/a
13	BioMedical Engineering OnLine	14	120	78.57	1.89
14	Biomedical Signal Processing and Control	14	58	57.14	2.67
15	2015 37th Annual International Conference of the IEEE Engineering in Medicine and Biology Society (EMBC)	14	67	92.86	n/a
16	2006 28th Annual International Conference of the IEEE Engineering in Medicine and Biology Society, Vols 1–15	14	5	14.29	n/a
17	2012 Annual International Conference of the IEEE Engineering in Medicine and Biology Society (EMBC)	13	133	92.31	n/a
18	Biomedizinische Technik	13	78	92.31	n/a
19	2014 36th Annual International Conference of the IEEE Engineering in Medicine and Biology Society (EMBC)	13	67	84.62	n/a
20	2010 Annual International Conference of the IEEE Engineering in Medicine and Biology Society (EMBC)	12	53	83.33	n/a

ACPY, average citation per year; IF, impact factors.

**Table 4 micromachines-13-01225-t004:** Contribution of the top 8 authors in cuffless blood pressure research.

Author	TA	TPR%	TC	ACPP	h-Index	Institution
Zhang, Yuan-ting	40	1.91%	830	25.8	12	Chinese University of Hong Kong/Chinese Academy of Sciences
Mukkamala, Ramakrishna	24	1.15%	471	19.63	8	Michigan State University
Hahn, Jin-Oh	23	1.10%	458	19.91	8	University of Maryland College Park
Butlin, Mark	15	0.72%	70	4.67	5	Macquarie University
Sivaprakasam, Mohanasankar	15	0.72%	46	3.07	5	Indian Institute of Technology (IIT)-Madras
Avolio, Alberto P.	14	0.67%	67	4.79	5	Macquarie University
Poon, Carmen C. Y.	13	0.62%	280	21.54	5	Chinese University of Hong Kong
Nabeel, P. M.	13	0.62%	36	2.77	5	Indian Institute of Technology (IIT)-Madras

**Table 5 micromachines-13-01225-t005:** Top 10 most cited publications during the period of 1990–2020.

Rank	Author	Title	TC	TCY	Source	Year	Country
1	Allen et al. [53]	Photoplethysmography and Its Application in Clinical Physiological Measurement.	1327	94.79	Physiological Measurement	2007	England
2	Khan et al. [54]	Monitoring of Vital Signs with Flexible and Wearable Medical Devices.	405	81	Advanced Materials	2016	USA
3	Zheng et al. [46]	Unobtrusive Sensing and Wearable Devices for Health Informatics.	314	44.86	IEEE Transactions On Biomedical Engineering	2014	China
4	Tardy et al.	Noninvasive Estimate of the Mechanical-Properties of Peripheral Arteries from Ultrasonic and Photoplethysmographic Measurements.	247	8.23	Clinical Physics Additionally, Physiological Measurement	1991	Switzerland
5	Mukkamala et al. [20]	Toward Ubiquitous Blood Pressure Monitoring via Pulse Transit Time: Theory and Practice.	244	40.67	IEEE Transactions On Biomedical Engineering	2015	USA
6	Asada et al.	Mobile Monitoring with Wearable Photoplethysmographic Biosensors.	226	12.56	IEEE Engineering In Medicine Additionally, Biology Magazine	2003	USA
7	Chen et al. [27]	Continuous Estimation of Systolic Blood Pressure Using the Pulse Arrival Time and Intermittent Calibration.	202	9.62	Medical & Biological Engineering & Computing	2000	Japan
8	Pandian et al.	Smart Vest: Wearable Multi-Parameter Remote Physiological Monitoring System.	183	14.08	Medical Engineering & Physics	2008	India
9	Yilmaz et al. [55]	Detecting Vital Signs with Wearable Wireless Sensors.	168	15.27	Sensors	2010	England
10	Yilmaz et al. [56]	Flexible, Highly Sensitive, and Wearable Pressure and Strain Sensors with Graphene Porous Network Structure.	151	30.20	ACS Applied Materials & Interfaces	2016	China

**Table 6 micromachines-13-01225-t006:** Contribution of the top 20 research areas in cuffless blood pressure monitoring field.

Rank	WoS Research Area	TA	TPR(%)	TC	ACPP
1	Engineering, Biomedical	764	37.62	6485	8.49
2	Engineering, Electrical & Electronic	650	32.00	2617	4.03
3	Computer Science, Interdisciplinary Applications	188	9.26	2159	11.48
4	Medical Informatics	175	8.62	2304	13.17
5	Instruments & Instrumentation	152	7.48	1134	7.46
6	Computer Science, Information Systems	143	7.04	1264	8.84
7	Telecommunications	102	5.02	732	7.18
8	Computer Science, Theory & Methods	102	5.02	171	1.68
9	Mathematical & Computational Biology	93	4.58	1675	18.01
10	Physics, Applied	88	5.34	1164	13.23
11	Computer Science, Artificial Intelligence	86	5.34	272	3.16
12	Engineering, Multidisciplinary	61	4.58	257	4.21
13	Optics	56	4.43	113	2.02
14	Biophysics	53	2.61	347	6.55
15	Nanoscience & Nanotechnology	52	2.60	1242	23.88
16	Chemistry, Analytical	48	2.36	111	2.31
17	Materials Science, Multidisciplinary	45	2.22	480	10
18	Automation & control systems	38	1.87	1105	23.02
19	Computer Science, Hardware & Architecture	49	2.37	115	2.35
20	Acoustics	27	1.31	224	8.29

**Table 7 micromachines-13-01225-t007:** Reviews on topic of cuffless blood pressure monitoring.

Year	Author	Title	Source
2007	Allen et al.	Photoplethysmography and Its Application in Clinical Physiological Measurement	*Physiological Measurement*
2009	Sugawara et al.	Clinical Usefulness of Wave Intensity Analysis	*Medical & Biological Engineering & Computing*
2010	Yilmaz et al.	Detecting Vital Signs with Wearable Wireless Sensors	*Sensors*
2010	Wang et al.	Theory and Applications of the Harmonic Analysis of Arterial Pressure Pulse Waves	*Journal of Medical and Biological Engineering*
2010	Avolio et al.	Arterial Blood Pressure Measurement and Pulse Wave Analysis-Their Role in Enhancing Cardiovascular Assessment	*Physiological Measurement*
2011	Yamakoshi et al.	Current Status of Noninvasive Bioinstrumentation for Healthcare	*Sensors and materials*
2014	Peter et al.	A Review of Methods for Non-Invasive and Continuous Blood Pressure Monitoring: Pulse Transit Time Method Is Promising?	*IRBM*
2015	Pereira et al.	Novel Methods for Pulse Wave Velocity Measurement	*Journal of Medical and Biological Engineering*
2015	Buxi et al.	A Survey on Signals and Systems in Ambulatory Blood Pressure Monitoring Using Pulse Transit Time	*Physiological Measurement*
2015	Scherer et al.	Body-Monitoring and Health Supervision by Means of Optical Fiber-Based Sensing Systems in Medical Textiles	*Advanced Healthcare Materials*
2016	Gambarotta et al.	A Review of Methods for the Signal Quality Assessment to Improve Reliability of Heart Rate and Blood Pressures Derived Parameters	*Medical & Biological Engineering & Computing*
2016	Khan et al.	Monitoring of Vital Signs with Flexible and Wearable Medical Devices	*Advanced Materials*
2016	Benmira et al.	From Korotkoff and Marey to Automatic Non-Invasive Oscillometric Blood Pressure Measurement: Does Easiness Come with Reliability?	*Expert Review of Medical Devices*
2017	Gajdova et al.	Pulse Wave Analysis and Diabetes Mellitus. A Systematic Review	*Biomedical Papers-Olomouc*
2018	Arakawa et al.	Recent Research and Developing Trends of Wearable Sensors for Detecting Blood Pressure	*Sensors*
2018	Moraes et al.	Advances in Photopletysmography Signal Analysis for Biomedical Applications	*Sensors*
2018	Wang et al.	Towards a Continuous Non-Invasive Cuffless Blood Pressure Monitoring System Using PPG: Systems and Circuits Review	*IEEE Circuits and Systems Magazine*
2019	Rastegar et al.	Non-Invasive Continuous Blood Pressure Monitoring Systems: Current and Proposed Technology Issues and Challenges	*Australasian Physical & Engineering Sciences in Medicine*
2019	Harford et al.	Availability and Performance of Image-Based, Non-Contact Methods of Monitoring Heart Rate, Blood Pressure, Respiratory Rate, and Oxygen Saturation: A Systematic Review	*Physiological Measurement*
2019	Hong et al.	Wearable and Implantable Devices for Cardiovascular Healthcare: From Monitoring to Therapy Based on Flexible and Stretchable Electronics	*Advanced Functional Materials*
2019	Zhou et al.	A Review on Low-Dimensional Physics-Based Models of Systemic Arteries: Application to Estimation of Central Aortic Pressure	*Biomedical Engineering Online*
2019	Joshi et al.	Wearable Sensors to Improve Detection of Patient Deterioration	*Expert Review of Medical Devices*
2019	Stojanova et al.	Continuous Blood Pressure Monitoring as a Basis for Ambient Assisted Living (AAL)-Review of Methodologies and Devices	*Journal of Medical Systems*
2019	Senturk et al.	Towards Wearable Blood Pressure Measurement Systems from Biosignals: A Review	*Turkish Journal of Electrical Engineering and Computer Sciences*
2020	El-Hajj et al.	A Review of Machine Learning Techniques in Photoplethysmography for the Non-Invasive Cuff-Less Measurement of Blood Pressure	*Biomedical Signal Processing and Control*
2020	Rastegar et al.	Non-Invasive Continuous Blood Pressure Monitoring Systems: Current and Proposed Technology Issues and Challenges	*Physical and Engineering Sciences in Medicine*
2020	Yao et al.	Recent Progress on the Fabrication and Applications of Flexible Ferroelectric Devices	*Journal of Materials Chemistry C*
2020	Kim et al.	Printing Flexible and Hybrid Electronics for Human Skin and Eye-Interfaced Health Monitoring Systems	*Advanced Materials*

**Table 8 micromachines-13-01225-t008:** Technical contents of cuffless blood pressure monitoring related reviews.

Year	Methods	Signals Sources	Methods of BP Estimating Models
2007	PTT, VTT	PPG, ECG	Linear Regression
2009	PWV, Vascular impedance, Diastolic time index	PPW, ICG	\
2010	PWV	PPW	Regression
2014(1)	PTT, PWV	PPG, ECG, PCG, ICG, BCG	Linear regression, Stationary wavelet transform
2015(1)	PWV, Ultrasound,	PPG, PPW, MRI	
2015(2)	PTT, PAT	PPG, Bioimpedance, PCG	Regression (linear, nonlinear), PAT-BP coefficients
2016(1)		ECG, PPG	RVM
2018(1)	PTT, PAT, PWV	PPG	Regression, Machine learning (ML)
2018(2)	PWA, PTT	PPG, ECG	Regression
2018(3)	PTT,	PPG	Linear regression, BP transport model, ML(random forest, support vector, etc.)
2018(4)	PTT, PWV	PPG, ECG, PPW	Regression
2019(1)	PWA	PPG (Image based),	Bland–Altman analysis, Basic regression
2019(2)	PTT, PWA	ECG, PPG	Regression (Linear and nonlinear), ML(BP), SVM, DNN
2019(3)	PTT, PWV	ECG, PPG, BCG	Regression, ANN, DNN
2019(4)	PTT, Ultrasonic	ECG, PPW (Based Soft Electronic)	Regression
2020(1)	PWA	PPW,	Regression, Machine learning
2020(2)	PTT, PWV, PAT, PWA, Ultrasound	PPG, ECG, BCG, SCG,	Regression, ML (SVM. BP, etc.) DNN (LSTM, RNN), ANN
2020(3)	PTT, PWV, PAT, PWA	PPG, ECG	Regression, DNN(RNN), ML (SVM, BP, Regression tree, etc.)

## Data Availability

Restrictions apply to the availability of these data. Data was obtained from Web of Science and are available at https://www.webofscience.com (accessed on 22 July 2022) with the permission of Web of Science.

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
