# Peer review of "Cuffless Blood Pressure Monitoring: Academic Insights and Perspectives Analysis"

_micromachines, 2022, doi:10.3390/mi13081225_

Round 1
Reviewer 1 Report
The article content is more bibliographic than scientific. There are very few scientific issues presented and any of them related to the specific of the journal “Micromachines”. It could be interesting for the scientific community how many articles were published and from which country, however, a review should be focused on materials, technologies, methods rather than this type of statistics.
I sincerely understand the volume of work conducted and efforts made by authors but from a review I would expect more scientific work and the focus should be on the state of art in this area in order to provide to the scientists working in this domain insights on the work conducted by their colleagues.
Author Response
Thank you for your suggestion. We have added some references for the " Micromachines " journals (Page 2).
Reviewer 2 Report
Review MDPI Micromachines
Title: Non-Invasive Wearable Blood Pressure Monitoring: Academic 2 Insights and Perspectives Analysis
The authors employed a bibliometric analysis of publications in the field of wearable blood pressure monitoring systems. The study highlights which countries, institutions, authors, and journals have had the most impact in the field so far. The main contribution of the manuscript is the application of bibliometric methods to quantify wearable blood pressure monitoring research activities in the world. This kind of information might be useful for the scientific community, but the manuscript could use some modifications. I suggest acceptance of the manuscript with revisions.
First of all, the English language has to be reviewed in Introduction and Methods. As a non-native English speaker, I understand the difficulty of writing in English and I do not expect a native level from everyone. But, the Introduction is really hard to understand because of the English level, which in not acceptable for publication. Please review. For the rest of the paper, I would say the level of English is acceptable, but would gain from language polishing.
The Introduction lists many different papers and the methods used for estimating blood pressure. However, results of those study should be added to the introduction so that the reader can understand better the implication of those methods and how it influenced the scientific community.
On line 42, the description of oscillometric method should be changed. It uses a cuff, not a balloon. Please see the following paper for a more accurate definition:
A. Chandrasekhar et al., “Formulas to Explain Popular Oscillometric Blood Pressure Estimation Algorithms,” Front. Physiol., vol. 10, 2019.
The paragraph on line 114 gives a brief definition of bibliometrics technology. However, it does not give a good understanding of the technology for a reader that is new to this. The definition should be enhanced so that the reader can better understand the contribution of this manuscript.
The introduction should have a section describing other review papers and how this manuscript is different from those. I can understand by the end of the paper that the contribution is about the bibliometrics, but this should be clearly stated in the Introduction.
On line 92, PPT is used instead of PTT. This is observed throughout the text. Please revise.
The first paragraph of Method is hard to understand. Please revised so that the contribution of the manuscript can be better understood.
On line 144, one keyword is “Cufless”. It should be “Cuffless” or “Cuff-less”.
The lines on Figure 1 should be on 2 different graphs. The variations in the blue one is hardly visible.
On line 195, The expression Latin countries is used. It is not clear as to what they are. Please also define what is relatively low ACPP.
In Table 1, Spain is all in capitalized letters. Please correct.
On line 228, it is written that “the research quality is also relatively impressive”. This is based on what criteria? Also, relatively impressive is not clear. Relative to what?
In tables in general, there are columns using only capitalized letters. Please change this. It will make the tables more readable. Do so for the journals name listed in the text.
On line 264, you say “Mukkamala R from India”. From India is irrelevant. Please remove.
In Table 5, some of the papers are hardly about blood pressure measurement technologies. Can you please add comments about this in the manuscript? For instance, Rank 1 has only a section on blood pressure. So, most of the citations might not be about the blood pressure portions of the article. Also, the titles in Table 5 are difficult to read as they all read continuously. Maybe put a dot at the end of the titles and change the alignment. I think ref 9 has an incomplete title.
The section 4.2.1 is speculative and, also, some statements are not true. It is true that PTT/PAT are still the base for BP measurement. However, PWA is probably a bigger field of research nowadays and this is unrelated to flexible electronics. This is usually performed using only one PPG sensor. Also, ANN is not an automatic feature detection method and has nothing to do with time and/or frequency domain. ANN can learn features from data, but is does not extract them. Also, when you say the “the process of machine learning in pressure pulse wave and ultrasonic…”, does not mean much. Please add some references and describe the idea you had in mind.
In limitations, please discuss the limitation of bibliometrics in and of itself. Then discuss the further limitations when applied to your dataset.
On line 418, you say “countries… East Asia and Europe”. These are not countries. Please reformulate.
On line 430, I don’t think we can assume that the distribution of articles in many journals indicates that the topics has widely concerned journals. I would argue that it probably means the opposite; that it was difficult to publish on the topic.
I’m not sure about the use of “non-invasive wearable” in the title and in the text in general. I think cuffless would be more appropriate. For instance, an oscillometric cuff monitoring system can be wearable and is non-invasive. Also, I would argue that if it is wearable, then it is non-invasive.

Author Response
Reviewer #2:
The authors employed a bibliometric analysis of publications in the field of wearable blood pressure monitoring systems. The study highlights which countries, institutions, authors, and journals have had the most impact in the field so far. The main contribution of the manuscript is the application of bibliometric methods to quantify wearable blood pressure monitoring research activities in the world. This kind of information might be useful for the scientific community, but the manuscript could use some modifications. I suggest acceptance of the manuscript with revisions.
- First of all, the English language has to be reviewed in Introduction and Methods. As a non-native English speaker, I understand the difficulty of writing in English and I do not expect a native level from everyone. But, the Introduction is really hard to understand because of the English level, which in not acceptable for publication. Please review. For the rest of the paper, I would say the level of English is acceptable, but would gain from language polishing.
Reply: Thank you. The contribution of this paper is about the bibliometrics. We have revised the intro part and restated the sentences that were not easy to understand.
- The Introduction lists many different papers and the methods used for estimating blood pressure. However, results of those study should be added to the introduction so that the reader can understand better the implication of those methods and how it influenced the scientific community.
Reply: Thank you. To enable the reader to better understand the implications of these methods and its impact on the scientific community, we have added the results of each study (Page 1-3).
- On line 42, the description of oscillometric method should be changed. It uses a cuff, not a balloon. Please see the following paper for a more accurate definition:
- Chandrasekhar et al., “Formulas to Explain Popular Oscillometric Blood Pressure Estimation Algorithms,” Front. Physiol., vol. 10, 2019.
Reply: Thank you. We have corrected this error and changed the relevant references (Page 1 and Page 19).
- The paragraph on line 114 gives a brief definition of bibliometrics technology. However, it does not give a good understanding of the technology for a reader that is new to this. The definition should be enhanced so that the reader can better understand the contribution of this manuscript.
Reply: Thank you. The definition has been changed to “a way to study the structure, quantitative relationship, change pattern and variable management of literature information by using mathematics, statistics and other measurement methods to explore the structure, characteristics and pattern of the literature system and bibliometric characteristics as the object of study”(Page 3).
- The introduction should have a section describing other review papers and how this manuscript is different from those. I can understand by the end of the paper that the contribution is about the bibliometrics, but this should be clearly stated in the Introduction.
Reply: The last paragraph of the introduction illustrates the difference between this review and other reviews (page 4).
- On line 92, PPT is used instead of PTT. This is observed throughout the text. Please revise.
Reply: We have fixed this error (Page 2 and Page 4).
- The first paragraph of Method is hard to understand. Please revised so that the contribution of the manuscript can be better understood.
Reply: Thank you. We have modified the expression of this paragraph (Page 3).
- On line 144, one keyword is “Cufless”. It should be “Cuffless” or “Cuff-less”.
Reply: We have corrected this error (Page 3).
- The lines on Figure 1 should be on 2 different graphs. The variations in the blue one is hardly visible.
Reply:In this diagram, the point is not to show specific values, but to show the trend of the two curves.
- On line 195, The expression Latin countries is used. It is not clear as to what they are. Please also define what is relatively low ACPP.
Reply: We have deleted this ambiguous sentence.
- In Table 1, Spain is all in capitalized letters. Please correct.
Reply: We have corrected this error (Table 1).
- On line 228, it is written that “the research quality is also relatively impressive”. This is based on what criteria? Also, relatively impressive is not clear. Relative to what?
Reply: We believe that the quality of the papers can be reflected by the citation rate. The total number of citations of the Chinese University of Hong Kong and the Chinese Academy of Sciences is 1305 and 902, which is more respectable compared to the total number of citations of the University of London in the UK. And the h-index of the two universities also exceeds that of the University of London.
- In tables in general, there are columns using only capitalized letters. Please change this. It will make the tables more readable. Do so for the journals name listed in the text.
Reply: We have fixed this error (Table2, Table 3 and Table 7).
- On line 264, you say “Mukkamala R from India”. From India is irrelevant. Please remove.
Reply: We have removed “from India”(Page 9).
- In Table 5, some of the papers are hardly about blood pressure measurement technologies. Can you please add comments about this in the manuscript? For instance, Rank 1 has only a section on blood pressure. So, most of the citations might not be about the blood pressure portions of the article. Also, the titles in Table 5 are difficult to read as they all read continuously. Maybe put a dot at the end of the titles and change the alignment. I think ref 9 has an incomplete title.
Reply: We have distilled the highlights of the literature related to blood pressure measurement in Table 5(Page 11-12). We have added dots after the headings in Table 5 and changed the line spacing of Table 5 to 25 mm to read the headings clearly.
- The section 4.2.1 is speculative and, also, some statements are not true. It is true that PTT/PAT are still the base for BP measurement. However, PWA is probably a bigger field of research nowadays and this is unrelated to flexible electronics. This is usually performed using only one PPG sensor. Also, ANN is not an automatic feature detection method and has nothing to do with time and/or frequency domain. ANN can learn features from data, but is does not extract them. Also, when you say the “the process of machine learning in pressure pulse wave and ultrasonic…”, does not mean much. Please add some references and describe the idea you had in mind.
Reply: We have revised our paper (Page 8).
- In limitations, please discuss the limitation of bibliometrics in and of itself. Then discuss the further limitations when applied to your dataset.
Reply: Thank you. We have added bibliometric restrictions in the limitation.
- On line 418, you say “countries… East Asia and Europe”. These are not countries. Please reformulate.
Reply : We have changed the “countries” to the “regions”.
- On line 430, I don’t think we can assume that the distribution of articles in many journals indicates that the topics has widely concerned journals. I would argue that it probably means the opposite; that it was difficult to publish on the topic.
Reply: We have made changes in the text (Page 18).
- I’m not sure about the use of “non-invasive wearable” in the title and in the text in general. I think cuffless would be more appropriate. For instance, an oscillometric cuff monitoring system can be wearable and is non-invasive. Also, I would argue that if it is wearable, then it is non-invasive.
Reply: Thank you. We have changed “non-invasive wearable” to “cuffless” in the text.
Reviewer 3 Report
This is a very well-crafted review article that provides a good idea of ​​the state of the issue of non-invasive wearable blood pressure monitoring.
In the Introduction part, I would welcome a more detailed introduction to the issue, including a picture of the pressure curve with the definition of systolic and diastolic pressure. Similarly, it would be appropriate to present the definition of PAT and PTT graphically.
On line 43, I would recommend replacing the word "a baloon" with "a cuff". On line 44, replace "oscilloscope method" with "oscillometric method". Line 92 says PPT - should probably be PTT or PPG.
In the Methods section, the keyword on line 144 is "cufless" - it should be "cuffless".
In the listed results, I did not find a section or mention of validation studies for devices according to (AAMI, ESH, etc.). Whether with a positive results, e.g.: Bilo G., Zorzi C., Ochoa Munera J.E., Torasco C., Giuli V. and Parati G. Validation of the Somnotouch-NIBP non-invasive continuous blood pressure monitor according to the European Society of Hypertension International Protocol revision 2010. Blood Pressure Monitoring 2015 Oct; 20(5):291-94 doi: 10.1097/MBP.0000000000000124. Or even with a negative validation result.
I would highly recommend supplementing this review article with this section, including the validation protocols.
Perhaps it would be appropriate to mention clinically validated devices that are already available on the market (e.g. SomnoTOUCH NIBP).
From this point of view, it would be appropriate to add to the Conclusions section that although the accuracy of the devices is questionable, there are first already clinically proven devices for use in clinical practice.
In my opinion, the bubble charts are harder to read, including the descriptions on the axes - it would probably be better to enlarge the images.
I would also recommend a general typographical check of the text.
Thank you.
Author Response
Reviewer #3:
This is a very well-crafted review article that provides a good idea of ​​the state of the issue of non-invasive wearable blood pressure monitoring.
- In the Introduction part, I would welcome a more detailed introduction to the issue, including a picture of the pressure curve with the definition of systolic and diastolic pressure. Similarly, it would be appropriate to present the definition of PAT and PTT graphically.
Reply: We have added these figures (Page 2).
- On line 43, I would recommend replacing the word "a baloon" with "a cuff". On line 44, replace "oscilloscope method" with "oscillometric method". Line 92 says PPT - should probably be PTT or PPG.
Reply: Thank you. We have fixed this error (Page 1 and Page 2).
- In the Methods section, the keyword on line 144 is "cufless" - it should be "cuffless".
Reply: Thank you. We have corrected this error (Page 3).
- In the listed results, I did not find a section or mention of validation studies for devices according to (AAMI, ESH, etc.). Whether with a positive results, e.g.: Bilo G., Zorzi C., Ochoa Munera J.E., Torasco C., Giuli V. and Parati G. Validation of the Somnotouch-NIBP non-invasive continuous blood pressure monitor according to the European Society of Hypertension International Protocol revision 2010. Blood Pressure Monitoring 2015 Oct; 20(5):291-94 doi: 10.1097/MBP.0000000000000124. Or even with a negative validation result.
I would highly recommend supplementing this review article with this section, including the validation protocols.
Reply: Thank you. We have supplemented this reference (page 4).
- Perhaps it would be appropriate to mention clinically validated devices that are already available on the market (e.g. SomnoTOUCH NIBP).
Reply: Thank you. We have added this point (Page 2). This shows that blood pressure measuring equipment has begun to be used in the current medical system. For one thing, it can make the article more convincing. Another point is to better reflect the value of the article.
- From this point of view, it would be appropriate to add to the Conclusions section that although the accuracy of the devices is questionable, there are first already clinically proven devices for use in clinical practice.
Reply: Thank you very much, We have added this point(Page 2). This can prove the application prospect of blood pressure detection technology in the medical field and demonstrate the significance of this paper.
- In my opinion, the bubble charts are harder to read, including the descriptions on the axes - it would probably be better to enlarge the images.
Reply: We are very sorry about this and we have enlarged the image to make it easier for readers to read.
- I would also recommend a general typographical check of the text.
Reply: Thank you. We have checked the full text.

Round 2
Reviewer 2 Report
All my comments have been addressed. The English language could be improved, but it's not a limitation to understand the manuscript.
Author Response
Thank you very much for your suggestion. We have made some improvements to the English language of the manuscript.
Reviewer 3 Report
Thank you for commenting on individual comments and supplementing the article.
On line 46, the inaccurate term "oscilloscope method" still remained - please use "oscillometric method".
For Figure 2, I would prefer to use a figure that makes it clear what PTT and PAT are (e.g. https://www.researchgate.net/publication/281439102_A_System_for_In-Ear_Pulse_Wave_Measurements/figures?lo=1 and https://www.ahajournals .org/doi/10.1161/HYPERTENSIONAHA.121.17891).
Thank you.
Author Response
- On line 46, the inaccurate term "oscilloscope method" still remained - please use "oscillometric method".
Reply: Thank you. We have changed that term on line 46 (page 2).
- For Figure 2, I would prefer to use a figure that makes it clear what PTT and PAT are (e.g. https://www.researchgate.net/publication/281439102_A_System_for_In-Ear_Pulse_Wave_Measurements/figures?lo=1 and https://www.ahajournals .org/doi/10.1161/HYPERTENSIONAHA.121.17891).
Reply: Thank you. We have replaced Figure 2 (page 2).